# Health Promoting School Interventions in Latin America: A Systematic Review Protocol on the Dimensions of the RE-AIM Framework

**DOI:** 10.3390/ijerph17155558

**Published:** 2020-07-31

**Authors:** Patrícia de Oliveira Bastos, Ana Suelen Pedroza Cavalcante, Wallingson Michael Gonçalves Pereira, Victor Hugo Santos de Castro, Antonio Rodrigues Ferreira Júnior, Paulo Henrique Guerra, Kelly Samara da Silva, Maria Rocineide Ferreira da Silva, Valter Cordeiro Barbosa Filho

**Affiliations:** 1Post-graduate Program in Collective Health, Ceara State University, Fortaleza 60714-903, Brazil; paty.bastos@aluno.uece.br (P.d.O.B.); anasuelen15@hotmail.com (A.S.P.C.); wallingsonmichaelgp@hotmail.com (W.M.G.P.); vsantosdecastro@gmail.com (V.H.S.d.C.); arodrigues.junior@uece.br (A.R.F.J.); rocineide.ferreira@uece.br (M.R.F.d.S.); 2Federal University of Fronteira Sul, Chapeco 89815-899, Brazil; paulo.guerra@uffs.edu.br; 3Research Center for Physical Activity and Health, Federal University of Santa Catarina, Florianopolis 88040-900, Brazil; kelly.samara@ufsc.br; 4Federal Institute of Education, Science and Technology of Ceara, Aracati 62800-000, Brazil

**Keywords:** health promotion at school, school health, systematic review, Latin America, policy making, implementation science, school age populations, program evaluation

## Abstract

Understanding the dimensions of internal and external validities (e.g., using the RE-AIM model: Reach, Effectiveness/Efficacy, Adoption, Implementation, and Maintenance) of school interventions is important to guide research and practice in this context. The aim of this systematic review protocol is to synthesize evidence on the RE-AIM dimensions in interventions based on the Health Promoting School (HPS) approach from the World Health Organization (WHO) in Latin America. Studies of interventions based on HPS-WHO that were carried out in Latin America involving the population of 5 to 18-year-olds will be eligible. Searches in nine electronic databases, a study repository, the gray literature, and the retrieved articles’ reference lists will be performed, without year or publication language limits. Study selection and data extraction will be conducted by independent researchers. Data on intervention implementation will be summarized in categories of HPS-WHO actions: (1) school curriculum, (2) changes in the social and/or physical environment of schools, and (3) actions with families and the community. A previously validated tool will be used to summarize the information on the dimensions of the RE-AIM model. The strengths and limitations of the included studies will be evaluated using the Critical Appraisal Skills Program (CASP) tool, and the confidence level of evidence will be assessed according to the GRADE CERQual tool.

## 1. Introduction

Health promotion in childhood and adolescence has a significant impact on health, society, and the economy [1,2]. Globally, more than 1.1 million deaths due to preventable or treatable causes [3] are estimated among adolescents aged between 10 and 19 years old (more than 3000 per day). The countries of Latin America and the Caribbean are characterized by being the most unequal region in the world in terms of income distribution [4]. This matrix of social inequality results in health inequities, as it directly influences the choice of quality food, housing conditions, risk behaviors and access to health services [5]. Latin American and Caribbean countries stand out on the global scenario with the highest child homicide rates and on worrying estimates of risk factors for the development of chronic non-communicable diseases [2,5,6]. Therefore, confrontation of health harms and health promotion for the young population of this region is imperative in order to decrease these health inequities.

The school has been recognized as an appropriate context for consolidating actions in health promotion and confronting social vulnerabilities suffered in childhood and adolescence [7,8,9]. In 1997, the World Health Organization (WHO) issued the technical report “Promoting Health Through Schools”, which recommends measures and political actions that allow the school to use its full potential to improve the health of children, adolescents, their families, school staff, and the community [10].

Thus, the WHO’s Health Promoting Schools (HPS-WHO) proposal includes a wide school health care model that goes beyond the biomedical perspectives and traditional education [10]. In this perspective, school health intervention strategies should seek planning with a view to education and to health, promoting flexibility in curricula in a participative manner that focuses on meeting students’ well-being and health needs in their totality [7,11,12].

Dimensions related to internal validity (which refers to how true the results of the study are for the studied population) and external validity (aimed at extending the results to other representatives of the population of interest) are important to elucidate the potential impact on public health of populational-scale intervention [13,14]. In this regard, the RE-AIM model (Reach, Effectiveness/Efficacy, Adoption, Implementation, and Maintenance) aims to determine if the dimensions can assist in the evaluation of the potential impact on public health interventions and in the creation and implementation of policies and programs on a large-scale [15,16,17]. One of the main dimensions of the RE-AIM model for stakeholders, police-makers, and practitioners is the implementation [18,19]. Elements such as acceptability, appropriateness, feasibility, fidelity, cost, and penetration are necessary for decision-making on whether or not to carry out new treatments, practices, and health services [20,21]. Knowing these elements can assist administrators in carrying out these actions on a large scale and in analyzing whether these elements have increased the chances of success in the school population’s health [21]. Thus, the RE-AIM model is favorable because it allows a balance between the elements of internal and external validity of interventions at both the organizational and the individual levels [17,22,23], thus its widespread use worldwide [17].

Previous reports have summarized the relevant information about interventions based on HPS-WHO [7,8,11,24,25]. Two studies have synthesized data on the HPS-WHO and showed that interventions promote positive changes in different scholars’ health indicators, such as nutritional status, physical fitness, physical activity, fruit and vegetable intake, tobacco use, and bullying [7,11]. However, only two of the 67 interventions analyzed were carried out in a Latin America context, and these were conducted in Mexico. Another systematic review summarized data of studies focused on the dimensions that may facilitate or hinder the process of implementing interventions based on HPS-WHO worldwide; however, none have been conducted in the region in question [11]. Finally, a systematic review that synthesized studies on HPS-WHO in Latin America from 1996 to 2009 identified only eight studies, but only two of them had experimental designs [9,25].

Considering these gaps, studies in countries and contexts of social, economic, and political vulnerabilities that impact young population’s health need to be encouraged and summarized [1,2,5,6,10,25,26]. In particular, Latin American countries have peculiar political, economic, and social contexts that are different from countries in other regions in which interventions based on HPS-WHO are carried out (mainly North America and Europe), which fact may influence both children and adolescents’ health and how policies and interventions regarding school health promotion are implemented. A synthesis of evidence with methodological rigor and specific data from these countries may guide the different actors involved in their schools (politicians, managers, teachers, family members, and scholars) in the development and implementation of viable strategies for health promotion in Latin American schools.

Therefore, this is a systematic review protocol that will evaluate whether the dimensions of internal and external validities (based on the RE-AIM model) are addressed in interventions based on the HPS-WHO in Latin America. A secondary aim of this protocol is to synthesize information (qualitative and quantitative) on the implementation process of the selected interventions.

## 2. Materials and Methods

### 2.1. Protocol and Registration

This systematic review was based on the methodological guidelines for evaluation of implementation studies proposed by Cochrane [27,28]. This protocol was registered on the PROSPERO platform (International Prospective Register of Systematic Reviews), under registration number CRD42020168069, with its writing following the guidelines of the Systematic Review and Meta-analysis Protocols (PRISMA-P) (Appendix A) [29].

### 2.2. Elegibility Criteria

Eligibility criteria were organized based on the items suggested by the SPIDER strategy (a version of the PICO strategy adapted for qualitative/mixed-method studies, based on Sample, Phenomenon of Interest, Design, Evaluation, and Research type) [30].

Sample: School-aged children and adolescents (aged 5 to 18 years old or the mean age in this age group) who are students of schools located in Latin American countries. Studies regarding exclusively school staff (for example, teachers) and other members of the school community will not be included.Phenomenon of Interest: Potential studies must have conducted one or more actions in each of the three dimensions of the HPS-WHO model: (1) school curriculum, (2) changes in the social and/or physical environment of schools, and (3) actions with families and the community [11]. Due to the plurality of possible results, the presence of the term or direct reference to the WHO report will not be required [7]. In addition, strategies should focus on a component of health or well-being (for example, mental health, healthy lifestyle, sexual health, oral health, hygiene, vaccination, substance use, and multi-component interventions) [7].Evaluation: Studies will be included if they present information on one of the aims of this protocol: (1) internal and external validity (RE-AIM model) and (2) evaluation of the implementation process:Studies will be considered whether they reported information will include at least one of the five dimensions that represent the RE-AIM model [23,31]. The reach (R) will represent the absolute number, proportion, and representativeness of individuals who are willing to participate in a given intervention or initiative compared to those who have given up or who have not joined. The effectiveness/efficacy (E) will represent the impact or repercussion of an intervention on important outcomes, whether negative, positive, or financial. Adoption (A) will represent the absolute number, proportion, and representativeness of the agents or organizations that are willing to join the program. Implementation (I) will be at the individual level when the agents use the intervention strategies themselves. At the organizational level, it will refer to the loyalty of the action agents to the various stages of an intervention protocol. Finally, maintenance (M) will represent the long-term beneficial effects at the individual level. At the organizational level, it will represent these effects as a program becomes institutionalized over time, thus being part of the routine, practice, or local policy [15,23,31].The implementation process will consider the taxonomy of Proctor et al. [21]: acceptability, adoption, appropriateness, feasibility, fidelity, implementation cost, penetration, and sustainability. These elements can be measured by evaluating subjects in the intervention plan (children, teachers, school staff, parents, and others), through qualitative, mixed or quantitative methods to measure the results related to the process (if planning or training was carried out, what resources were used, amount, and reach), people/agents (if there was fidelity to the intervention proposal, what adaptations were necessary for each type of context) and products (if objectives were achieved) [20,21].Research type: Finally, amidst the wide variety of study designs for evaluating the implementation of policies, programs, or practices [19], controlled intervention studies will be included, without necessarily requiring randomization for allocation into groups. They will be included regardless of whether data collection and analysis processes were based on quantitative, qualitative, or mixed methods.

### 2.3. Information Sources

No limits regarding year, language, or publication status will be applied, provided that they will attend the eligibility criteria.

Searches for potential articles started in May 2020 and will be updated if more than 12 months pass before the publication. The following platforms will be searched:Electronic databases (*n* = 9): Medline, PubMed, LILACS, Web of Science, Scopus, PsycINFO, Eric, SciELO, and Cochrane Library;Gray literature (*n* = 5): Brazilian Digital Library of Theses and Dissertations (BTDT) (http://bdtd.ibict.br/vufind/), Networked Digital Library of Theses and Dissertations (NDLTD) (http://www.ndltd.org/), International Clinical Trials Registry Platform (ICTRP) (http://apps.who.int/trialsearch/), Brazilian Clinical Trials Registry (REBEC) (http://www.ensaiosclinicos.gov.br/) and Google Scholar (screening of the first 200 results).Complementary strategies: Search in specialized websites on the theme (www.iuhpe.org; www.ashaweb.org; www.who.int; www.cdc.gov; www.unesco.org; www.freshschools.org; www.hbsc.org; and www.paho.org). Complementary searches will include the screening of the titles in the references of the chosen articles and literature reviews on the theme [7,8,11,22,25,26,32].

### 2.4. Search Strategy

In order to produce the best results from the search strategy, the terms were tested and defined based on the DeCS (Health Sciences Descriptors) and MeSH (Medical Subject Headings) of the National Library of Medicine, which were aligned with text words from the scientific literature pertinent to the area, as explained in Appendix A The search strategy was developed with terms related to the population of interest (children and adolescents from Latin American countries), schools, school health interventions, and experimental studies. Terms were defined in consensus meetings between researchers, including researchers with experience in searching electronic databases, and with the support of reviews on child and adolescent health [1,2], Health Promoting School [10,11,24], and school-based interventions [7]. The Boolean connectors (“OR”, “AND”, and “NOT”) were arranged according to the characteristics and search guidance of each database. Searching in all databases will be performed using terms in English, and databases originated from Brazil (LILACS, SciELO, BTDT, and REBEC) will be searched in Portuguese complementarily. Search strategies were developed acording to the recommendations of the Peer Review of Electronic Search Strategies (PRESS) Statement [33].

### 2.5. Study Selection

All results will be exported to EndNote Web, and one of the authors (P.B.) will remove any duplicates. Two reviewers (P.B. and A.S) will independently analyze the titles and abstracts of each of the results, which will be selected based on the inclusion criteria. Articles that do not meet these criteria will be excluded. Any discrepancies between the two reviewers regarding the included studies will be discussed; then, consensus will be reached by a third author (V.C.).

After this first search, the full texts will be read independently by the two authors (P.B. and A.S). Disagreements will again be discussed with a third author (V.C.) to reach a consensus. All studies excluded at this stage will be listed and included in the “Table of Excluded Studies”, with statements about the rationale for exclusion. For selected texts that are not available in full, that are unpublished, or that are ongoing studies, the authors will contact the authors to request any necessary information. The PRISMA flow chart will be used to report the study selection process.

### 2.6. Data Extraction

Data will be extracted by one reviewer and verified by the other (P.B. and A.S.) using a standardized data extraction spreadsheet. Disagreements during this process will be resolved during a consensus meeting involving a third author (V.C.). Authors of unpublished and ongoing studies will be contacted to request any necessary information. Each study included will receive an identifier consisting of the authors’ names and the primary reference year. Publications representing the same study will be combined into a single identifier. For studies with the same authors and the same years, letters in alphabetical order will be added as a third identifier.

Data related to the implementation of HPS-WHO interventions in Latin American schools will be extracted according to the form contained in Appendix A. This form was produced according to guidance provided by Moore et al. [20] and Proctor et al. [21] on the evaluation of the implementation of data extraction.

Data regarding intervention studies’ internal and external validity will be extracted. Then, a checklist (Appendix A) will be produced according to the RE-AIM model and adapted by Brito et al. [31] for use in systematic reviews, with data about the characterization of the studies added. In total, the checklist will consist of 21 items, which will be related to each segment of the RE-AIM model: Reach (five items), Effectiveness/efficacy (four items), Adoption (six items), Implementation (three items), and Maintenance (three items) [31].

### 2.7. Assessment of Risk of Bias

The Critical Assessment Skills Program (CASP) quality checklist will be used to assess the risk of bias in the included studies, considering the possible distinctions between the study designs (e.g., randomized controlled trial, cluster randomized controlled trial, and nonrandomized controlled trial) [34]. This tool has been recommended for evidence synthesis that address complex interventions and the evaluation of implementation process [28].

CASP assesses studies’ risk of bias based on 10 questions related to the following domains: objectives, methodology, research design, recruitment strategy, data collection, rigor of data analysis, reflexivity-related issues, ethical issues, discoveries, and contribution to the research. The possible answers to these ten questions are yes, no, unclear, or not applicable.

Two independent reviewers (P.B. and A.S.) will critically assess the risk of bias in studies referred to in the synthesis. A third reviewer (V.C.) will be consulted for resolution of doubts, agreement or consensus. No study will be excluded based on assessment of the risk of bias; on the contrary, the methodological rigor of each study will be considered for the confidence assessments of each finding in the review. Nevertheless, the strengths and methodological limitations of the studies will be discussed among the authors until they reach a consensus. This will be done based on each item and its impact on the main inferences from the studies and the review. In other words, the use of the total scores or the classification of the methodological quality of the included studies will not be applied [28,34].

### 2.8. Data Synthesis

A descriptive synthesis will summarize information about the methodological and general characteristics of the included studies (for example, year and country of implementation, population group, and intervention strategies).

For the synthesis of evidence on the dimensions of internal and external validity, the scores of the 21-item checklist for each study will be calculated, considering the general and specific score of the domain. The overall quality of the studies will also be assessed and determined based on the reports’ frequency of meeting the 21 items related to the RE-AIM framework. To that end, criteria proposed by Brito et al. [31] will be applied in order to classify the overall score as low (0 to 7 points), moderate (8 to 14 points), or high (15 or higher) quality.

The results of the implementation process will be presented according to the following dimensions: acceptability, adoption, appropriateness, feasibility, fidelity, implementation cost, penetration, and sustainability [20,21]. Quantitative information will be organized in tables and presented in a narrative synthesis (considering a preliminary heterogeneity of data). Qualitative data will be organized using the IRAMUTEQ (R INTERFACE for multidimensional analysis of texts and questionnaires) software in different stages: similarity analysis, word cloud, classic textual statistics, research on group specificities, and descending hierarchical classification. This categorization will be discussed and defined by two researchers (P.B. and A.S.) during the meetings.

Finally, the level of confidence of each finding in the qualitative summaries of the implementation process’s dimensions will be analyzed using GRADE CERQual, © Lewin et al., Oslo, Norway [35]. This tool provides a systematic and transparent framework for assessing the level of confidence in the conclusions of a review based on the following components: (1) methodological limitations, (2) coherence of the review finding, (3) adequacy of data supporting a review finding, and (4) relevance of the included studies for the review question. Four levels are used to describe the overall assessment of confidence: high, moderate, low, and very low, considering the recommendations for this tool. Two researchers will perform this process (P.B. and A.S.) in a consensus meeting to validate the coding.

Potential publication biases will be analyzed narratively. For example, the authors will assess whether information about implementation or scores in the RE-AIM dimensions will be different according to the publication status (journal articles vs. unpublished documents), period, and language of publication. This protocol is exempt from ethical approval since it will be a review of previously published articles. Ethical aspects of the included studies will be reported and discussed based on the CASP domain on ethical issues [34].

## 3. Final Considerations

The HPS-WHO approach has been recognized as a suitable model in interventions aimed at combining health and education in an integral way in schools [1,7,9,10,25]. Considering the potentiality and particularities of this model in Latin America, we propose this systematic review in order to synthesize the level and quality of evidence on intervention of the HPS model in these countries. By emphasizing the implementation process, we can expose which elements are essential for the assertiveness of future interventions or even political decisions about the student’s health. We expect that the clarity of the evidence on the process of implementing these interventions will provide relevant information for the formulation of interventions and public policies to be widely implemented in schools in Latin America. The main components that favor or hinder its implementation in this context will also be addressed.

Moreover, the evidence-based decision making on health policies and programs in schools in in Latin American countries may consider the results of this protocol because it will summarize dimensions of internal (e.g., whether HPS-WHO intervention may impact students’ health behaviors) and external (e.g., whether HPS-WHO are feasible and suitable for the schools from different areas) validities. Notwithstanding, knowing the quality of the study reports on the elements of internal and external validities in interventions (based on the RE-AIM approach) will indicate which elements need improvement in terms of description and scientific evaluation. This will guide future research and interventions aimed at students’ health in Latin America.

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
