# Peer review of "Health Promoting School Interventions in Latin America: A Systematic Review Protocol on the Dimensions of the RE-AIM Framework"

_ijerph, 2020, doi:10.3390/ijerph17155558_

Round 1

Reviewer 1 Report

Thank you for submitting this manuscript, thereby providing me with the opportunity to review your proposed work.  This manuscript aims to outline a rigorous desk-based research protocol which could offer novel insights into current knowledge of school-based health promoting interventions in Latin America.

I have made a number of observations below for your consideration.

Specific comments

Can the manuscript clarify to the reader what is meant by the internal and external validities in the context of the topic of this systematic review.

Introduction:

Could some more information be added regarding the specific health inequalities for Latin America.  This could be for the global region or specific countries within the Americas.  This helps to give greater context and to help support why this SR is needed.

Lines 87-91 – please take another look at this sentence.  I am not sure “fact” is needed, or you might want to consider presenting as two sentences. 

Methods: 

Have you conducted an initial search to scope the literature?  I ask this because the feasibility of the proposed methods will be impacted according to number of published articles returned.  May the search need to be time bounded or limited in other ways?  Have the authors’ considered this and can this be outlined in this manuscript.

Please consider, and include, a justification for the use of the SPIDER strategy as opposed to a more widely used alternative (PICO, PICOS, PICOT).  Will the proposed synthesis be qualitative, quantitative or mixed methods, and has this impacted upon the decision to adopt the SPIDER strategy)?

Similarly, can the choice of risk of bias assessment tool (i.e., CASP) be justified.  Why were alternative risk of bias tools deemed not appropriate for the proposed review?  And, will the CASP qualitative checklist be used?

Please consider the ethics of systematic reviews and include ethics in the protocol outlined.  Could ethics form part of the eligibility criteria?

Line 260 – typographical/spelling error “documents”

Final Considerations:

Can a brief analysis of the strengths of the proposed protocol be provided, and, furthermore, what this means for commenting upon internal and external validities of interventions.

Author Response

We are grateful for all the attention devoted to this study. All observations were discussed among the authors and were substantial for our learning construction, which resulted in an even more rigorous and systematic protocol, with adjustments in all sections of the protocol. Please find our answers according to the suggestions in the attached file. 

Regards,

The authors

Reviewer 2 Report

Review for IJERPH-849837

This manuscript aimed to synthesize evidence on the dimensions that represent processes of implementing interventions based on the Health Promoting School (HPS) approach from the World Health Organization (WHO) in Latin America and its internal and external validity. This paper used the RE-AIM (Reach, Effectiveness/Efficacy, Adoption, Implementation, and Maintenance) model and others tools to address the confidence level of evidence. The paper is generally well written but there are some typos and sentences throughout that need to be corrected. While the paper has some strengths, it has some limitations that I would like to share with the co-authors.

The main goal is not totally clear. Are the authors proposing to evaluate the process of implement interventions or specific issues on the internal and external validity? I would suggest the authors to remove “internal and external validity” in the (main) aim and keep as a step of RE-AIM.

Introduction: line 86-87. The word “summarized” is typed twice.

I firmly believe that the arguments to conduct the manuscript are NOT a strong justification for conducting research. The authors pointed out that the gaps of the previous literature are: (1) gaps in research and practices toward understanding the internal and external validity (2) Fewer included studies conducted in Latin-America (3) Examination of a few elements that underpin decisions on implementation and on the internal and external validity. Are there few studies conducted in Latin America in previous review due the lack of study quality? Or was there an increase of studies in Latin America reporting elements of internal and external validity in the past decades? I think the main point here is to clarify what will be inclusion criteria (2.5 Study selection). I´m supposing that any intervention study conducted in Latina America based on HPS framework will be included, regardless if they had all internal and external information. I would recommend authors to emphasize the importance of their research to better understanding the elements of internal and external validity, and how that will support translating research into practice. In other words, the arguments should be intrinsic related to how the proposed research can move forward the public health area.

While searching the gray literature is an important step forward, it is necessary include the others specific countries digital library. Or, clarify if only Brazil has one.

When the authors say “no limits of language”, what does specifically means? Other languages than English, Portuguese and Spanish will be use, or considered? Do the authors believe that will appear any other language?

Considering the context of this review, I would suggest the authors include as a supplementary files the search terms in Portuguese and Spanish language as well.

Why the authors choose not included behaviors, such as nutrition, physical activity, sexual behaviors, and etc, as a search terms?  In addition, I would suggest include “health risk behavior” as search term.  

Author Response

(The authors gave the same response as above.)
